# Urea Fertilization Significantly Promotes Nitrous Oxide Emissions from Agricultural Soils and Is Attributed to the Short-Term Suppression of Nitrite-Oxidizing Bacteria during Urea Hydrolysis

**DOI:** 10.3390/microorganisms12040685

**Published:** 2024-03-28

**Authors:** Yiming Jiang, Yueyue Zhu, Weitie Lin, Jianfei Luo

**Affiliations:** 1School of Biology and Biological Engineering, South China University of Technology, Guangzhou 510006, China; 13450208234@163.com (Y.J.); bio-z@foxmail.com (Y.Z.); 2Guangdong Key Laboratory of Fermentation and Enzyme Engineering, South China University of Technology, Guangzhou 510006, China; 3MOE Joint International Research Laboratory of Synthetic Biology and Medicine, South China University of Technology, Guangzhou 510006, China

**Keywords:** N_2_O emission, urea fertilization, NOB suppression, nitrite accumulation

## Abstract

The application of urea in agricultural soil significantly boosts nitrous oxide (N_2_O) emissions. However, the reason for nitrite accumulation, the period of nitrite-oxidizing bacteria (NOB) suppression, and the main NOB species for nitrite removal behind urea fertilization have not been thoroughly investigated. In this study, four laboratory microcosm experiments were conducted to simulate urea fertilization in agricultural soils. We found that within 36 h of urea application, nitrite oxidation lagged behind ammonia oxidation, leading to nitrite accumulation and increased N_2_O emissions. However, after 36 h, NOB activity recovered and then removed nitrite, leading to reduced N_2_O emissions. Urea use resulted in an N_2_O emission rate tenfold higher than ammonium. During incubation, *Nitrobacter*-affiliated NOB growth decreased initially but increased later with urea use, while *Nitrospira*-affiliated NOB appeared unaffected. Chlorate suppression of NOB lasted longer, increasing N_2_O emissions. Urease inhibitors effectively reduced N_2_O emissions by slowing urea hydrolysis and limiting free ammonia production, preventing short-term NOB suppression. In summary, short-term NOB suppression during urea hydrolysis played a crucial role in increasing N_2_O emissions from agricultural soils. These findings revealed the reasons behind the surge in N_2_O emissions caused by extensive urea application and provided guidance for reducing N_2_O emissions in agricultural production processes.

## 1. Introduction

Nitrous oxide (N_2_O), a greenhouse gas known for its extended atmospheric lifespan of approximately 116 years, has experienced a notable surge in concentration, reaching 332.1 parts per billion (ppb) in 2019; notably, the concentration has been rising at a rate of 0.95 ppb per year over the past ten years [1,2]. Among the sources of increased N_2_O emissions, agricultural soils are the primary contributor (accounting for 65%), especially after the widespread use of synthetic fertilizers [3,4,5]. According to the report by the Intergovernmental Panel on Climate Change (IPCC), it was estimated that approximately 1% of the nitrogen input was converted into N_2_O and emitted directly from the soil, with this emission factor (EF) contributing significantly to greenhouse gas emissions [6,7]. However, when ammonium or urea were utilized as nitrogen fertilizers, the EF value exhibited a substantial increase that surpassed linear growth (1%), particularly evident when the nitrogen inputs surpassed 200 kg N ha^−1^ [8,9,10,11]. Urea is the most commonly used nitrogen fertilizer in agriculture, accounting for more than 70% of the global nitrogen utilization [12]. In China, the amount of fertilized urea among different crops was about 45 to 183 kg ha^−1^, which contributed to the amount of greenhouse gas (GHG) emissions ranging between 149 and 607 kg CO_2_-eq ha^−1^, accounting for 60% of the total GHG emissions from agriculture [13].

In contrast to the fertilization of ammonium, amended urea often contributes to more N_2_O emissions from agricultural soils [14,15,16]. During urea hydrolysis, the generated free ammonia (FA) is supposed to suppress the activity of nitrite-oxidizing bacteria (NOB) and then reduce the consumption of nitrite, which could serve as a proximal substrate for N_2_O production [16,17,18]. However, the exact mechanism and pattern behind how urea hydrolysis promotes N_2_O production have not been thoroughly studied. Thus, we propose the following questions: (1) Nitrite concentration during microcosm experiments is usually below the detection limit, so is it nitrite accumulation that leads to N_2_O emissions? (2) The concentration of the generated FA is very low, so does FA suppress NOB activity? (3) NOB suppression results in nitrite accumulation and the promotion of N_2_O emissions, but how long would this last? (4) The genera *Nitrobacter* and *Nitrospira* are two common NOBs that are distributed in soils, but which kind is responsible for nitrite removal and is sensitive to FA stress?

In this study, to answer these questions, we conducted four microcosm experiments to simulate urea fertilization in agricultural soils. The concentration changes of N-species and N_2_O emissions during short-term (72 h) and long-term (9 days) incubations were tracked, and the time-series changes of *Nitrobacter* and *Nitrospira* were quantified. In addition, a urease inhibitor and an NOB inhibitor were applied to retard urea hydrolysis and suppress NOB activity, respectively, and then we assessed their influences on nitrite accumulation and N_2_O emissions. The nitrite-oxidizing potential changes of four different soils after exposure to FA stress were also evaluated.

## 2. Materials and Methods

### 2.1. Soil Collection

Soils for the microcosm experiments were sourced from farmland located in Guangzhou, Guangdong Province (coordinates: 22°55′39.31″ N, 113°26′38.40″ E and 22°55′45.55″ N, 113°26′41.35″ E), in February 2022. Characterized by a subtropical monsoon climate, this area experiences an average yearly precipitation of 1420.9 mm and a mean air temperature of 23.0 °C. Agricultural activities in this region have been ongoing for over two decades—specifically, the cultivation of vegetables and fruits. To obtain the soil samples, five cores were randomly collected at a depth of 5–20 cm and subsequently combined to form a single sample. Following transportation to the laboratory and air-drying, the samples underwent sieving to eliminate stones and plant debris, ensuring a uniform particle size of less than 2 mm. Measurements of the solanum soil properties revealed a bulk density of 1.26 g cm^−3^, a pH level of 6.83, a moisture content of 23.3% (*w*/*w*), and a total organic carbon content of 1.1%. Additionally, the extractable inorganic nitrogen content was found to be 80.20 µg N g^−1^ soil_dw_, with NH_4_^+^-N at 0.74 µg g^−1^ soil_dw_ and NO_3_^−^-N at 79.46 µg g^−1^ soil_dw_. The nitrite concentration was below the detection limit. The detailed properties of other soil samples can be found in Appendix A.

### 2.2. Experimental Setup

Microcosm experiments were conducted following the methodology outlined by Hink [19]. In summary, approximately 12 g of soil (corresponding to 10 g of dry-weight soil) was placed into 120 mL serum bottles. After being sealed with butyl rubber stoppers and metal crimp tops, the bottles were incubated in the dark at a temperature of 30 °C. To stabilize the microbial community and prevent any pulse effect on soil respiration, a 10-day pre-incubation period was applied. To ensure aerobic conditions, the bottles were periodically opened and resealed every 3 days during the pre-incubation time.

Following the soil pre-incubation period, four distinct microcosm experiments were conducted. Notably, the solanum soils served as the basis for the first three experiments, whereas the fourth experiment utilized four separate soil samples. (1) The first microcosm was used to assess the effect of urea fertilization on soil N_2_O emissions within a short incubation time and clarify that the nitrite-oxidation activity was only briefly suppressed during urea hydrolysis. In this microcosm, the soils were amended with 250 µg N g^−1^ soil_dw_ of ammonium or urea; the soil and gas samples were obtained by sacrificial sampling after the incubation for 0, 12, 24, 36, 48, and 72 h, and a total of 30 serum bottles (1 soil × 1 concentration × 2 fertilizers × 3 replicates × 5 samplings) were used. (2) The second microcosm was used to assess the effect of different fertilizers on soil N_2_O emissions during a 9-day incubation. In this microcosm, the soils were amended with 100, 250, or 450 µg N g^−1^ soil_dw_ of ammonium or urea; the samples of soil and headspace gas were obtained by sacrificial sampling after the incubation for 0, 3, 6, and 9 days, and a total of 72 serum bottles (1 soil × 3 concentrations × 2 fertilizers × 3 replicates × 4 samplings) were used. (3) The third microcosm was used to demonstrate that the inhibition of NOB activity was able to increase soil N_2_O emissions and the reduction of urea hydrolysis was able to reduce soil N_2_O emissions. In this microcosm, the soils were amended with 250 µg N g^−1^ soil_dw_ of ammonium and 10/30 µg g^−1^ soil_dw_ of potassium chlorate as an NOB inhibitor, or they were amended with 250 µg N g^−1^ soil_dw_ of urea and 6.4 / 12.8 µg g^−1^ soil_dw_ of N-(n-butyl) thiophosphoric triamide (NBPT, aladdin, Shanghai, China) as a urease inhibitor. The soil and gas samples were obtained by sacrificial sampling after the incubation for 0, 3, 6, and 9 days, and a total of 48 serum bottles (1 soil × 2 inhibitors × 2 inhibitor concentrations × 3 replicates × 4 samplings) were used. (4) The fourth microcosm was used to prove that short-term exposure to FA could suppress the NOB activity in soils. The ammonia gas, with a final content of 0.9, 2.7, and 4.5% (*v*/*v*), was injected into the sealed serum bottles using a syringe; meanwhile, the acetylene with a final content of 0.01% (*v*/*v*) was also injected into bottles, in order to suppress the ammonia oxidation to nitrite. The soil samples were obtained by sacrificial sampling after the incubation for 0, 12, 24, and 48 h and used for the determination of nitrite-oxidizing potential (NOP); a total of 192 serum bottles (4 soils × (3 concentrations + 1 control) × 3 replicates × 4 samplings) were used. The initial moisture content of all microcosms was adjusted to 30% (*w*/*w*).

### 2.3. Chemical Determination and Analysis

Before the determination of nitrogen species, soil samples were mixed thoroughly and subjected to extraction by 20% (*w*/*v*) of calcium sulfate solution (2 g L^−1^). The concentrations of NO_3_^−^ in the extractions were measured using ion chromatography with a Dionex IonPac AS16 column from Thermo Fisher Scientific (Waltham, MA, USA). Specifically, the Griess–Ilovay method was employed for NO_2_^−^ determination, while the Indophenol Blue method (ISO/TS, 2003) was used for NH_4_^+^ quantification. The pH detector (Hach, Loveland, CO, USA) was utilized to ascertain the pH of the soil. Before the determination of soil pH, soil samples were subjected to homogenization in 20% (*w*/*v*) of potassium chloride solution (1 mol L^−1^). Gas samples (3 mL) were taken using a gas syringe and used for the measurement of N_2_O. The concentration of N_2_O in the headspace was measured using a gas chromatography system (GC9790plus, manufactured by FULI in Zhejiang, Wenling, China) that was equipped with a 63Ni electron capture detector. Quantification of N_2_O was achieved by comparing the peak areas obtained from the sample with those of a reference gas (supplied by Messer, Foshan, China). A five-point standard calibration curve (0.98, 9.99, 24.89, 50.03, 100.03 ppm) with an R^2^ value greater than 0.997 was used for the accurate calculation of N_2_O concentration. The mean nitrite-oxidation rate (NOR) of soils was analyzed according to Equation (1):(1)$$NOR=\frac{C_{{NO}_{3(t2)}^{-}}-C_{{NO}_{3(t1)}^{-}}}{t2-t1}$$

The mean ammonia oxidation rate (AOR) of soils was analyzed according to Equation (2):(2)$$AOR=\frac{C_{{NO}_{3(t2)}^{-}}+C_{{NO}_{2(t2)}^{-}}-C_{{NO}_{3\left(t1\right)}^{-}}-C_{{NO}_{2(t1)}^{-}}}{t2-t1}$$

In Equations (1) and (2), $C_{{NO}_{3(t2)}^{-}}$ and $C_{{NO}_{2(t2)}^{-}}$, or $C_{{NO}_{2(t2)}^{-}}$ and $C_{{NO}_{2(t1)}^{-}}$, mean the concentration of NO_3_^−^ or NO_2_^−^ at two different incubation times.

Nitrite-oxidation potential (NOP) was determined by a modified method according to previous studies [20,21]. Briefly, 1.0 g soil was obtained and inoculated into 25 mL of N1 culture medium [22], which contained 0.5 g L^−1^ KCl, 0.485 g L^−1^ MgSO_4_·7H_2_O, 1.0 g L^−1^ NaCl, 0.3 g L^−1^ KH_2_PO_4_, 0.1 g L^−1^ CaCl_2_·2H_2_O, 1.0 g L^−1^ CaCO_3_, 0.21 g L^−1^ NaHCO_3_, 0.69 g L^−1^ NaNO_3_, 0.2 g L^−1^ Na_2_EDTA, and 0.1% (*v*/*v*) trace element solution (2.1 g L^−1^ FeSO_4_·7H_2_O, 0.062 g L^−1^ H_3_BO_3_, 0.017 g L^−1^ CuCl_2_·2H_2_O, 0.1 g L^−1^ MnC1_2_·4H_2_O, 0.036 g L^−1^ Na_2_MoO_4_·2H_2_O, 0.07 g L^−1^ ZnCl_2_, 0.19 g L^−1^ CoCl_2_·6H_2_O, and 0.024 g L^−1^ NiCl_2_·6H_2_O). Cultivations were incubated in a shaker with a speed of 150 rpm at 30 °C. After the cultivation for 0, 3, 6, 9, and 12 h, the culture samples were obtained and used for the nitrite concentration determination. The NOP was analyzed according to Equation (3):(3)$$NOP=\frac{k_{p}\times V}{1+w\%}$$

In Equation (3), $k_{p}$ (µmol L^−1^ h^−1^) means the NOR calculated from the linear decrease in NO_2_^−^ concentration; V means the volume of 25 mL; 1 means the quality of 1.0 g soil; and w% means the 30% of soil moisture content. The unit of NOP is µmol g^−1^ soil_dw_ h^−1^.

The residual nitrite-oxidation potential (RNOP) of soils was analyzed according to Equation (4):(4)$$RNOP=\frac{{NOP}_{t}}{{NOP}_{ck}}$$

In Equation (4), ${NOP}_{t}$ means NOP at different incubation times, ${NOP}_{ck}$ means NOP of the control check. The unit of RNOP is %.

### 2.4. DNA Extraction and qPCR Analysis

Genomic DNA in soils was extracted using the EZNA Soil DNA Kit (Omega Bio-Tek, Norcross, GA, USA), according to the manufacturer’s instructions. DNA concentration was quantified using a NanoDrop spectrophotometer (Thermo Fisher Scientific, Waltham, MA, USA) and verified by agarose gel electrophoresis. The *nxrA* (specifically targeted to the *Nitrobacter*) and *nxrB* (specifically targeted to the *Nitrospira*) genes were quantified by qPCR using primer pairs F1norA/R2norA and nxrB169f/nxrB638r [16], respectively. The PCR reaction was carried out in a total volume of 20 µL, consisting of 10.0 µL of TransStar Tip Green qPCR SuperMix sourced from TransGen Biotech (Beijing, China), 0.6 µM of each primer, 0.4 µL of a passive reference dye (50× concentration), and 2 µL of 10× diluted DNA. The reaction was conducted using the Applied Biosystems 7500 Real-time PCR system (Applied Biosystems, Forster City, CA, USA). Standard curves were prepared using six serial tenfold dilutions ranging from 10^2^ to 10^7^ gene copies mL^−1^. Agarose gel electrophoresis and melting curves were used to guarantee the specificity of the amplification products. The R^2^ values for all standard curves were >0.99, and the primer efficiencies ranged from 95% to 103%.

### 2.5. Statistical Analysis

Results are expressed as means and standard errors (*n* = 3). A factorial two-way ANOVA was employed to examine the impact of both substrate and incubation duration on the abundances of *nxrA* and *nxrB* genes. Duncan tests were used to assess significant differences in means. The effect of NO_2_^−^ concentration on N_2_O emission rate was tested using linear regression analysis. The effect of incubation time on residual nitrite-oxidizing potential was tested by linear regression analysis. The linear fit was analyzed using Origin 2017 (OriginLab, Northampton, MA, USA). All statistical computations were carried out using SPSS 21.0 software (IBM Corporation, Armonk, NY, USA).

## 3. Results

### 3.1. Urea as Nitrogen Fertilizer Short-Termly Suppressed the Soil Nitrite-Oxidizing Activity

During the first microcosm, the nitrite was always detectable in the soils with the addition of urea, with concentrations ranging from 2.71 to 25.97 µg N g^−1^ soil_dw_ (Figure 1b), whereas it remained barely detectable in those treated with ammonium (Figure 1a). Due to the NOR being lower than the AOR during the first 36 h, the concentration of nitrite was continuously accumulated and reached the maximal value (25.97 µg N g^−1^ soil_dw_) after the incubation for 36 h (Figure 1b); after this time, the nitrite concentration gradually decreased because the NOR was higher than the AOR. It is worth noting that the AOR in urea-fertilized soils was always higher than that in ammonium-fertilized soils during the short-term incubation (*p* < 0.05); however, the NOR was lower during the first 36 h but much higher during the following 36 h (Figure 1b).

The N_2_O was found to be almost linearly produced along with the linear oxidation of ammonia in ammonium-fertilized soils (Figure 1a), while it was exponentially produced in urea-fertilized soils (Figure 1b). After 72 h incubation, the cumulative N_2_O production in urea-fertilized soils reached about 2271.5 ng N_2_O-N g^−1^ soil_dw_ (Figure 1b), which was 4.8 times higher than that in ammonium-fertilized soils (*p* < 0.001).

### 3.2. Urea as Nitrogen Fertilizer Promoted the Soil N_2_O Emission

During the second microcosm experiment, the N_2_O emission rate was determined in the range of 1.22–181.09 ng N g^−1^ soil_dw_ day^−1^ for soils when fertilized with 100–450 ng N g^−1^ soil_dw_ of ammonium (Figure 2a NH_4_^+^). However, when fertilized with the same concentration of urea, the maximal N_2_O emission rate was determined as 837.74 ng N g^−1^ soil_dw_ day^−1^, and the higher the concentration of urea that was supplied, the higher the rate of N_2_O emissions (Figure 2a Urea). The N_2_O emission rate was found to be closely related to the nitrite concentration that accumulated during the treatment (Appendix A, R^2^ > 0.89). For example, in soils supplemented with ammonium or a low concentration (100 µg N g^−1^ soil_dw_) of urea, no or only a very low concentration of nitrite was detected (Figure 2b NH_4_^+^), which is consistent with the low yield of N_2_O emissions (Figure 2a NH_4_^+^). However, when amended with 250 and 450 µg N g^−1^ soil_dw_ of urea, the nitrite concentration was determined as high as 19.13 and 16.78 µg N g^−1^ soil_dw_, respectively, on the third day, when the maximal N_2_O emission rate was observed. As the nitrite concentration decreased in the following days, the N_2_O emission rate simultaneously decreased (Figure 2). In a word, in contrast to the utilization of ammonium as a nitrogen fertilizer, urea contributed to more N_2_O emissions from soils, especially under high fertilization.

### 3.3. Urea as Nitrogen Fertilizer Short-Termly Suppressed the NOB Growth

The NOB growth and incubation time were determined by quantifying the number of *nxrA* and *nxrB* genes, which were specific for clades *Nitrobacter* and *Nitrospira*, respectively. As results shown in Figure 3, if we only look at the final results, the fertilization of ammonium or urea could benefit the growth of *Nitrobacter*-NOB (Figure 3a). However, upon examining the time-series changes of the *nxrA* gene, it was observed that ammonium fertilization consistently stimulated the growth of *Nitrobacter*-NOB (Figure 3a NH_4_^+^). Differently, urea fertilization initially hindered their growth during the first three days but subsequently facilitated continuous growth in the subsequent period (Figure 3a—Urea). Moreover, when growth inhibition occurred, the higher the concentration of urea supplied, the more inhibition was observed (Figure 3a Urea). Different from the growth of *Nitrobacter*-NOB, the *Nitrospira*-NOB was found to be promoted by low concentrations of nitrogen fertilizer but inhibited by high concentrations (Figure 3b). Except for the incubation under 450 µg N g^−1^ soil_dw_ of nitrogen fertilizers, the growth of *Nitrospira*-NOB had no very significant change (*p* > 0.05). It is obvious that the utilization of urea as a nitrogen fertilizer suppressed the growth of *Nitrobacter*-NOB in soils in the short term. Combining these results from the first and second microcosms, it is indicated that the utilization of urea as a nitrogen fertilizer suppressed NOB growth and activity in the short term, which in turn promoted nitrite accumulation and then N_2_O production.

### 3.4. Retarding Urea Hydrolysis or Maintaining NOB Activity Could Reduce N_2_O Emission

In the third microcosm experiments, chlorate and NBPT were utilized as they are commonly recognized as effective inhibitors of NOB and soil urease, respectively [23,24]. The addition of 10 or 30 µg g^−1^ soil_dw_ of KClO_3_ in soils was found to not only significantly suppress the NOR but also the AOR in both soils when fertilized with 250 µg N g^−1^ soil_dw_ of ammonium (Figure 4b). Due to the NOB suppression by 10 or 30 µg g^−1^ soil_dw_ of KClO_3_, the maximal concentration of nitrite was determined as 6.64 or 8.53 µg N g^−1^ soil_dw_ in soils; simultaneously, N_2_O emission rates as high as 671.48 or 778.44 ng N g^−1^ soil_dw_ day^−1^ were determined, respectively (Figure 4a). In contrast, when in the absence of an inhibitor (CK), no or very little (0.58 µg N g^−1^ soil_dw_) nitrite was detected in soils (Figure 4b), and the N_2_O emission rates were always at low levels (Figure 4a). As a consequence, NOB suppression was proved to be closely related to N_2_O production, and the more NOB was suppressed, the more N_2_O was produced.

After the addition of 6.4 or 12.8 µg g^−1^ soil_dw_ of NBPT, the period of urea hydrolysis was found to last for more than six days, which was much longer than in the soils without the addition of NBPT (less than three days) (Appendix A). Moreover, the nitrite was determined to be below the detection limit in all soils at any time (Figure 5b); the N_2_O emission rate was determined to be less than 100 ng N g^−1^ soil_dw_ day^−1^ (Figure 5a). Consequently, retarding the urea hydrolysis via the application of a soil urease inhibitor was able to reduce the N_2_O emissions from agricultural soils when using urea as a nitrogen fertilizer.

### 3.5. Free Ammonia Suppressed the NOB Activity in Soils

During the fourth microcosm, when incubated in the absence of FA (CK), the soil NOPs were all determined to have no significant change during the incubation time (Appendix A). However, when incubated in the presence of FA, the soil NOPs were found to decrease along with incubation time, and the more FA was added, the greater the reduction in NOP (Figure 6). The linear correlation analysis suggested that the soil NOPs were very closely related to the exposure time and FA concentration (the values of R^2^ ranged between 0.82 and 0.99). After 24 h, the RNOP remained at 61.92–92.98%, 44.24–80.34%, and 30.73–75.06% when incubated under 0.9%, 2.7%, and 4.5% of FA, respectively; after the incubation for 48 h, the RNOP remained at 42.49–94.17%, 18.66–75.10%, 10.85–52.30% (Appendix A). Among these four soils, the lettuce soil was found to be the most sensitive to FA stress (had the largest slope value); the solanum soil, which had been used to perform the four above microcosm experiments, was also sensitive to FA stress.

## 4. Discussion

### 4.1. The Suppression of NOB Disrupted the Equilibrium between Ammonia OXIDATION and Nitrite Oxidation, Ultimately Leading to an Elevation in Soil N_2_O Emission

Generally, nitrite accumulation in soils largely relies on the balance between ammonia oxidation and nitrite oxidation, and the accumulated nitrite is always found to be closely related to soil N_2_O emissions [14,25,26]. Due to the soil’s maximal NOR always exceeding the rate of nitrite production from ammonia oxidation, nitrite does not accumulate or remains below the detection limit [19,27,28,29]. In this study, when ammonium was used as a nitrogen fertilizer, the NOR was always equal to the AOR, which then led to no or a very small accumulation of nitrite (Figure 1a). However, when we used urea as a nitrogen fertilizer, the tendency of the balance to decrease in nitrite oxidation became more significant, which contributed to nitrite accumulation in soils (Figure 1b). If an NOB inhibitor was present, nitrite would also accumulate, even if using ammonium as a nitrogen fertilizer (Figure 4b). It is clear to see that NOB suppression broke the balance between ammonia and nitrite oxidation, which resulted in nitrite accumulation, and in turn, an increase in N_2_O emissions. Though heterotrophic denitrification is also a main source of soil N_2_O emissions, the N_2_O emission rate determined in this study was below 2.0 ng N g^−1^ soil_dw_ day^−1^ during the incubation (Appendix A), which only accounted for less than 1% of the total emissions. The accumulated nitrite is well-known to be responsible for N_2_O production in soils, but the exact mechanism of how nitrite accumulates is not clear enough. In this study, the balance between AOR and NOR was clearly demonstrated as the key factor in deciding whether nitrite would accumulate and soil N_2_O emissions would increase. This could be used as a kind of ecological indicator that is applied to evaluate the rationality and sustainability of fertilizer use and field management in agriculture.

### 4.2. NOB Suppression Only Occurred in a Short-Term Time during Urea Hydrolysis

The use of urea as a nitrogen fertilizer is well-known for its significant contribution to soil N_2_O emissions [12,13,14]. The FA generated from urea hydrolysis was supposed to suppress NOB activity and result in nitrite accumulation, then serving as a proximal substrate for N_2_O production [16,17,18]. However, the nitrite concentration was usually below the detection limit, which is confusing as we asked the question: does nitrite lead to N_2_O emissions? In the first microcosm experiment of this study, the NOR was found to be much lower than the AOR during the first 36 h when using urea as nitrogen fertilizer (Figure 1b), which resulted in large nitrite accumulation. After the incubation for 36 h, the NOB activity quickly recovered from the short-term suppression, resulting in a higher NOR compared to the AOR. Subsequently, this elevated NOR continuously removed the accumulated nitrite during the subsequent incubation time. In the second microcosm experiment, the nitrite concentration became very low and even below the detection limit after incubation for 3 days; in the meantime, the N_2_O emission rate was significantly reduced (Figure 2). Apparently, the accumulated nitrite contributed to the promotion of N_2_O emission; this was because the NOB suppression only occurred in the short term, and the accumulated nitrite subsequently promoted NOB growth (Figure 3a) and was rapidly removed.

The removal of NOB suppression during urea fertilization was due to the completion of urea hydrolysis. Because urea hydrolysis is often very rapid in soils (Appendix A), NOB suppression due to urea hydrolysis occurs in a short time. When urease inhibitors were present, the hydrolysis rate of urea decreased (Appendix A) and the ammonium produced by hydrolysis was rapidly consumed by ammonia-oxidizing microorganisms without inhibiting NOB. In this situation, no nitrite was generated, and the N_2_O emission rate was lower compared to that when using ammonium as a fertilizer (Figure 5a). If using KClO_3_ to suppress NOB, the N_2_O emission rate of the soils that used ammonium as a fertilizer was equal to that when using urea as a fertilizer; however, the suppression time could be for more than 6 days, which resulted in more N_2_O emissions (Figure 4a). It was once again suggested that the NOB suppression during urea hydrolysis only occurred in a short time.

In contrast to ammonium, fertilization with urea would contribute to more ammonia volatilization, especially under high pH levels or high input [30]. The FA that comes from volatilized ammonia is usually supposed to suppress NOB activity and then promote nitrite accumulation in soils [14,15,16]. The pH increase that resulted from urea hydrolysis is also reported to benefit nitrite accumulation; however, in the first microcosm of this study, the soil pH was determined to increase from 6.85 to 7.22 after 24 h incubation and gradually decrease in the following time (Appendix A). During the fourth microcosm, the addition of FA to the headspace of soils was proved to significantly suppress NOB activity even after short-term exposure (Figure 6). These results demonstrated that the FA generated from urea hydrolysis was mainly responsible for the short-term suppression of NOB activity and nitrite accumulation.

Urease inhibitors, such as NBPT, are usually used to reduce N_2_O emissions and NH_3_ volatilization from agricultural soils when using urea as a nitrogen fertilizer [31,32,33]. Due to the NOB suppression only occurring in a short time during urea hydrolysis, the inhibitor addition time should be simultaneous with the addition of urea; otherwise, it may not effectively reduce FA generation or could miss the best time to reduce N_2_O emissions.

### 4.3. Short-Term NOB Suppression Only Happened among the Genus Nitrobacter

The current known NOBs belong to the genera *Nitrobacter*, *Nitrotoga*, *Nitrococcus*, *Nitrospira*, *Nitrosopina*, *Nitrolancea*, and *Candidatus Nitromaritima* [34]. Among them, *Nitrobacter* and *Nitrospira* are usually identified to be dominant members in terrestrial environments, and they mainly contribute to nitrite oxidation in soils [16,35,36]. In this study, the initial abundance of *Nitrobacter* and *Nitrospira* were determined, respectively, as 2.61 × 10^6^ and 1.45 × 10^7^ copies g^−1^ soil_dw_ in soils (Figure 3a,b). During the incubation, the abundance of *Nitrobacter* continuously increased along with the ammonium input (Figure 3a NH_4_^+^), while it decreased at first and then increased later after the addition of urea (Figure 3a—Urea). Differently, the abundance of *Nitrospira* had no significant increase due to fertilizer input, but it significantly decreased after the addition of 450 µg N g^−1^ soil_dw_ of nitrogen fertilizer (Figure 3b). It seems that though the species of *Nitrospira* dominated in soils, they were insensitive to ammonium or urea. On the contrary, the *Nitrobacter* was very sensitive, especially under high inputs of urea; the pattern of their abundance changes was observed to be very consistent with the nitrite-oxidizing activity and the N_2_O emission rate. The abundance of *Nitrospira* is usually found to be higher than the *Nitrobacter* in soils [16,36,37]. However, *Nitrospira*-NOB has been found to exhibit a broader range of responses to urea fertilization, while *Nitrobacter*-NOB exhibits a ‘peak’ model of urea concentration [16]. As a result, it is likely that the *Nitrobacter*-affiliated NOB was primarily responsible for nitrite oxidation in the present soils when fertilized with ammonium or urea, and the short-term suppression by FA that generated during the process of urea hydrolysis only occurred to them. It is necessary to note that the NOB community including *Nitrobacter* and *Nitrospira* is usually shaped by soil character, fertilizer type, and land management. The pattern of NOB suppression, including the short time, would be last, and the main species responsible for nitrite oxidation will differentiate from soils.

## 5. Conclusions

This study demonstrated the exact patterns of urea fertilization in promoting soil N_2_O emissions. The main findings are presented as follows: (1) the accumulated nitrite was responsible for the increased N_2_O emissions; (2) the accumulated nitrite reached the maximal concentration after 36 h of incubation and then gradually reduced; (3) urea hydrolysis only resulted in short-term (within 36 h) NOB suppression, which simultaneously led to nitrite accumulation; (4) *Nitrobacter-* but not *Nitrospira*-affiliated NOB was suppressed during the urea hydrolysis; and (5) slowing down urea hydrolysis using a urease inhibitor could avoid short-term NOB suppression.

## Figures and Tables

**Figure 1 microorganisms-12-00685-f001:**
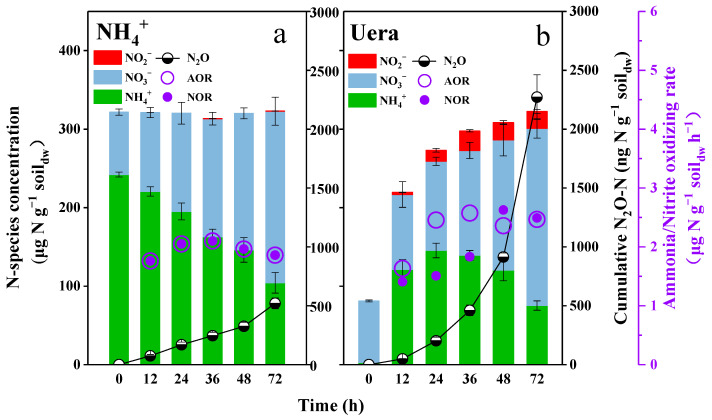
Short-term changes of cumulative N_2_O emission, N-species concentration, mean nitrite-oxidizing rate and mean ammonia-oxidizing rate during (**a**) ammonium or (**b**) urea fertilization. The soil pH changes during the microcosm experiment are shown in Appendix A.

**Figure 2 microorganisms-12-00685-f002:**
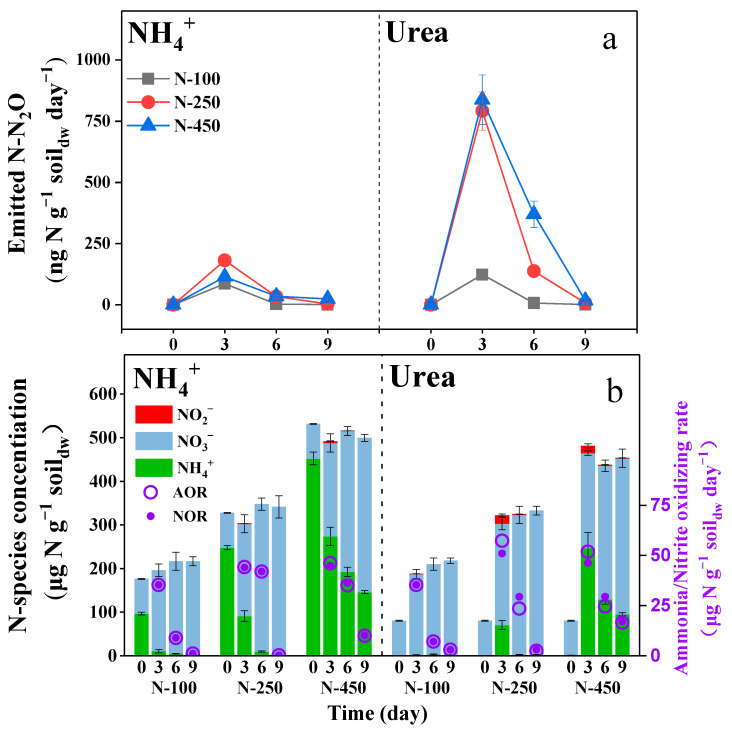
(**a**) Long-term changes of cumulative N_2_O emission, (**b**) Long-term changes of N-species concentration, mean nitrite-oxidizing rate and mean ammonia-oxidizing rate during ammonium or urea fertilization in soils. The soil pH changes during the microcosm experiment are shown in Appendix A.

**Figure 3 microorganisms-12-00685-f003:**
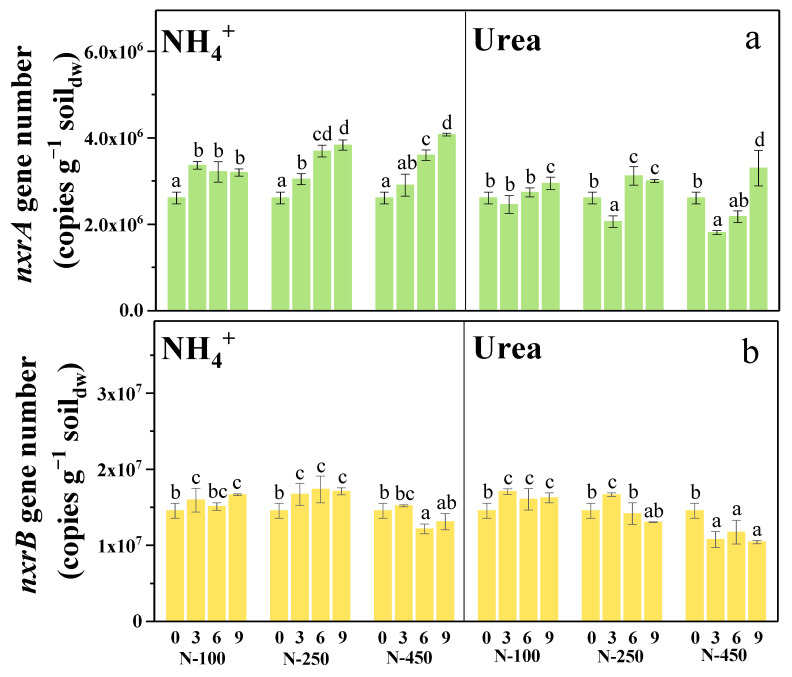
(**a**) Time-series changes of *nxrA*; (**b**) Time-series changes of *nxrB* gene number during ammonium or urea fertilization in soil sample. Different letters indicated significant differences (*p* < 0.05).

**Figure 4 microorganisms-12-00685-f004:**
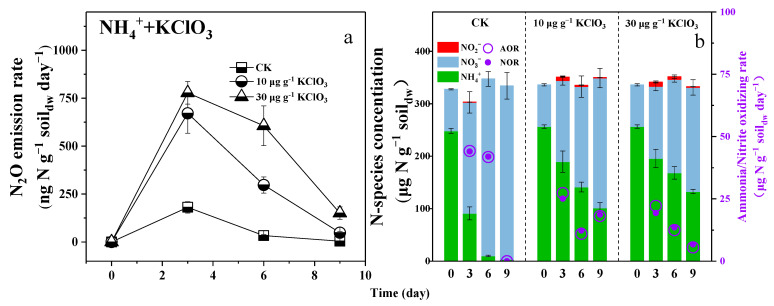
(**a**) Time-series changes of N_2_O emission rate; (**b**) Time-series changes of N-species concentration, mean nitrite-oxidizing rate and mean ammonia-oxidizing rate during ammonium fertilization in the presence of chlorate. CK: without addition of KClO_3_. The soil pH changes during the microcosm experiment are shown in Appendix A.

**Figure 5 microorganisms-12-00685-f005:**
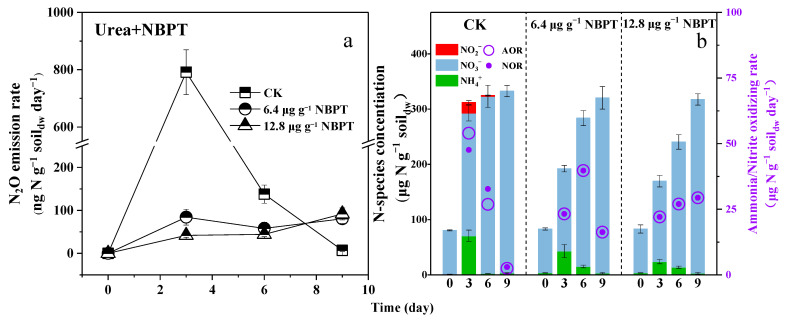
(**a**) Time-series changes of N_2_O emission rate; (**b**) Time-series changes of N-species concentration, mean nitrite-oxidizing rate and mean ammonia-oxidizing rate during urea fertilization in the presence of NBPT. CK: without addition of NBPT. The soil pH changes during the microcosm experiment are shown in Appendix A.

**Figure 6 microorganisms-12-00685-f006:**
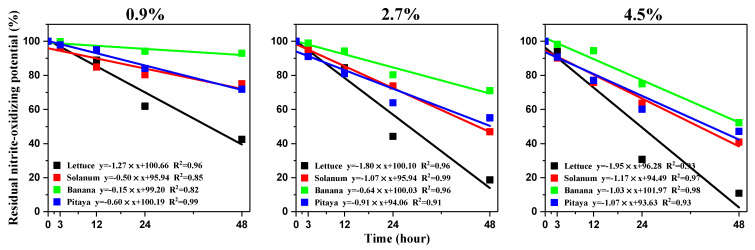
Linear correlation analysis between incubation time and residual nitrite-oxidizing potential in four soils when incubated with addition of 0.9, 2.7 or 4.5% FA.

## Data Availability

Data are contained within the article.

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
