# Peer review of "Urea Fertilization Significantly Promotes Nitrous Oxide Emissions from Agricultural Soils and Is Attributed to the Short-Term Suppression of Nitrite-Oxidizing Bacteria during Urea Hydrolysis"

_microorganisms, 2024, doi:10.3390/microorganisms12040685_

Round 1

Reviewer 1 Report

Comments and Suggestions for Authors

Thank you for the opportunity to evaluate the interesting research presented in the manuscript "The Significant Promotion of Urea Fertilization to Nitrous Oxide Emission from Agricultural Soils Attributed to the Short-Term Suppression of Nitrite-Oxidizing Bacteria during Urea Hydrolysis". The results obtained are very interesting. However, the manuscript requires corrections.

Detailed comments:

1.     Please complete the abstract briefly with the purpose, concept and methodology of the research. The obtained research results should not be presented so extensively in the abstract, please shorten them significantly. The abstract of a good journal article always ends with an outline of the benefits of the results and recommendations as a solution to the problem presented. The presented abstract lacks such information.

2.     Keywords should not repeat phrases contained in the title of the manuscript. Please change it.

3.     The authors made an interesting review of the literature. The current state of knowledge is clearly and precisely defined in the introduction. However, a hypothesis must be made. How does this work differ from existing literature? The last paragraph concluding the introductory part always emphasizes the innovative aspects of the experiments conducted with a clear goal and the significance of the obtained research results. The introduction lacks goals. In the closing part of the introduction, the authors presented conclusions. The aims of the experiments being conducted should be clearly and precisely presented to ensure a more comprehensive approach.

4.     The research was properly designed and conducted. The research methods were also correctly selected and described in detail, which allows the experiment to be reproduced. I have no objections. However, the statistical analysis used does not bring anything. I believe that an analysis should be carried out to search for a correlation between the nitrite oxidation potential and the parameters of the soils selected for the experiment (listed in Table S1). In order to determine the significance of the relationship between soil characteristics, non-parametric tests can be used: Spearman's rank correlation, t (Kendall) and g (Goodman and Kruskal). Especially since the authors performed the analysis in three repetitions. For statistically significant correlations of nitrite oxidation potential with soil parameters, the predictive capabilities of simple linear regression models (their statistical significance) can also be examined. Please complete the statistical analysis.

5.     The results of the conducted research were carefully analyzed and compared with the information available in the literature in the discussion section. The discussion can be expanded. All figures were correctly interpreted. However, the figures in the manuscript are blurry. Please correct it.

6. The conclusions presented are predictable. In a reputable journal, conclusions must be persuasive statements of what is considered novel, with strong support for the data and results presented. Why is your study important and how does it expand existing knowledge on the topic presented? It should be clearly noted what new contributions the research has made to the international literature and the final recommendations should be emphasized.

Reviewer 2 Report

Comments and Suggestions for Authors

The Significant Promotion of Urea Fertilization to Nitrous Oxide Emission from Agricultural Soils Attributed to the Short-Term Suppression of Nitrite-Oxidizing Bacteria during Urea Hydrolysis

Specific remarks:

Abstract:

Page 1, Line 20 – “NOB” Explain the meaning of this abbreviation.

Introduction:

Page 1, Line 41 – „IPCC“ When it is mentioned for the first time in the text, it is necessary to explain the abbreviation.

Materials and methods:

Page 2, Line 78 – „Table S1“ Explain all the methods used to obtain the results shown in Table S1.

Also, explain how the values in tables S2, S3 and S4 were obtained.

Page 3, Line 124 – „determination of soil pH“ How was the soil pH determined?

Results:

Page 5, Line 177 – “2.71 to 25.97 μg N g-1 soildw (Figure 1).” It would be good to label these Figures as 1a (NH4+) and 1b (Urea).

I assume these are the values for nitrites in Figure 1a. As can be seen from Figure 1a, the range of values you have written is large for what is shown in Figure 1a. If these values also refer to Figure 1b, I would ask the authors to explain each value obtained for NH4+ and Urea and indicate exactly which part of Figure 1 is explained. Otherwise, the explanation in the text with the representation of the Figure is a bit confusing.

Also, in Figure 1b, UERA is written instead of UREA.

Page 5, Line178 – “…AOR during the first 36 h…” indicate that you explain Figure 1b.

Page 5, Lines 203-208 - indicate that you explain Figure 2b.

Page 6, Line 219 – “12.6% to 58.9% (Figure 3a).” These values cannot be read from Figure 3a.

Page 7, Figure 4 - Give an explanation for the abbreviation CK

References

It is necessary to prepare a list of references according to the journal's instructions.

General remarks:

The paper is interesting and educational. A lot of work has been invested and accordingly I would like to praise the authors for that.

On the other hand, the explanation in the text with the representation of the Figures is a bit confusing. I am of the opinion that it is necessary to label the Figures better, or explain them better and more clearly in the text of the paper.

General opinion:

With proposed revisions, I suggest that the paper be considered for publication.

Best Regards,

Reviewer

Round 2

Reviewer 1 Report

Comments and Suggestions for Authors

Applies to manuscript: microorganisms-2930650-revised.

Comments:

1. The authors have slightly revised the abstract. However, they still have not defined the purpose of the research, concept and methodology. A brief mention of this topic should be included in the abstract. Furthermore, I don't understand the term "microcosmic experiments". I believe that it would be more accurate for the authors to specify the conditions in which they conducted their research. The abstract should clearly indicate whether these were laboratory or field experiments? I still think the abstract should be significantly improved.

2. Keywords have been corrected. I have no comments.

3. The introduction has only been slightly improved. The purpose of the research is still not clearly presented. The authors should indicate that the aim of the research was to determine: e.g. the causes of increased N2O emissions, the causes of nitrite accumulation, the type of microbiome responsible for nitrite removal. The sentences presented by the authors concluding the introductory part make the reader not fully understand the meaning of the experiments being conducted, additionally referred to by the strange term "microcosmic experiments". I believe that the final part of the introduction should be more precisely explained to the potential recipient.

4. The authors completed the statistical analysis of the research results, and the quality of the figures was improved. I have no further comments. However, please refer in the text to the statistical analysis performed, the results of which are illustrated in the figures.

5. The discussion of the obtained research results was not extended as requested in the previous evaluation of the manuscript. I still ask you to complete this chapter.

6. In their current form, the conclusions presented are acceptable. I have no comments.
